# Inspectable Tabular Foundation Models via In-Context Kernel Learning

**Ratmir Miftachov** [1] [*]   **Bruno Charron** [2]   **Simon Valentin** [3]

## Abstract

Tabular foundation models like TabPFN and TabICL achieve state-of-the-art performance through in-context learning, yet their architectures remain fundamentally opaque. We introduce KernelICL, a framework to enhance tabular foundation models with quantifiable sample-based inspectability. Building on the insight that in-context learning is akin to kernel regression, we make this mechanism explicit by replacing the final prediction layer with kernel functions (Gaussian, dot-product, kNN) so that every prediction is a transparent weighted average of training labels. We achieve prediction-layer inspectability through case-based reasoning, and quantify it via the perplexity of the weight distribution over retrieved training samples. On 55 TALENT benchmark datasets, KernelICL achieves performance on par with existing tabular foundation models, demonstrating that explicit kernel constraints on the final layer enable inspectable predictions without sacrificing performance.

## 1. Introduction

Tabular data remains the backbone of decision-making across industries, from healthcare diagnostics to financial risk assessment. While foundation models have revolutionized natural language processing and computer vision through in-context learning (ICL), their application to tabular data has only recently gained traction. Models like TabPFN (Hollmann et al., 2022; 2025) and TabICL (Qu et al., 2025) achieve state-of-the-art tabular classification through ICL in a single forward pass without dataset-specific tuning. TabPFN-2.5 (Grinsztajn et al., 2025) matches complex ensembles, establishing ICL as a strong paradigm for tabular learning. Recently, MITRA (Zhang

et al., 2025) has used diverse mixtures of synthetic priors to increase generalization and sample efficiency.

However, these models function as black boxes, limiting adoption in domains where decisions must be explainable. In healthcare, clinicians need to validate model reasoning against medical knowledge before acting on predictions (Rudin, 2019). In finance, regulatory frameworks increasingly demand transparency in automated decision-making (Bracke et al., 2019). Practitioners across domains require the ability to inspect which training cases inform a prediction and verify alignment with domain expertise. A comprehensive discussion of related work is provided in Appendix A.

Our goal of *prediction-layer inspectability*, is achieved by revealing the exact evidentiary support (training samples) driving the prediction output. Consider a practitioner evaluating a misclassified patient in the Pima Diabetes dataset (Table 1). With standard tabular foundation models like TabICL, the prediction is an opaque MLP output; even using post-hoc tools like SHAP, the practitioner only learns which features (e.g., high glucose) drove the decision. With KernelICL, the prediction is a transparent weighted average of the most similar training cases. The practitioner sees that the prediction is driven by patients with similarly high glucose levels, but notices the retrieved patients are much younger. This allows the practitioner to validate the model's logic, identify potential missing features, or override the prediction using domain expertise.

|  | Glucose | BMI | Age | BP | Insulin | Label |
|---|---|---|---|---|---|---|
| **Test Pt (Actual: ND)** | **194** | **26** | **67** | 80 | 0 | **Pred: D** |
| Neighbor 1 | 190 | 36 | 66 | 92 | 0 | D |
| Neighbor 2 | 197 | 26 | 39 | 74 | 0 | D |
| Neighbor 3 | 195 | 25 | 55 | 70 | 145 | D |

*Table 1.* Case-Based Reasoning in KernelICL (Pima Diabetes). A false positive (actual Non-Diabetic [ND], predicted Diabetic [D]) is explained by showing the practitioner the exact training cases (neighbors) driving the decision.

We introduce **KernelICL**, a framework to enhance tabular foundation models with quantifiable sample-based inspectability. Our main contributions are: 1) Prediction-layer inspectability; 2) efficient symmetric in-context embeddings; 3) systematic kernel function exploration, where symmetric embeddings enable distance-based kernels (Gaus-

---

[*]Work completed during an internship at Amazon. [1]Humboldt-Universität zu Berlin [2]Amazon [3]AWS AI Labs. Correspondence to: Ratmir Miftachov <contact@miftachov.com>, Bruno Charron <bcharro@amazon.fr>.

*Proceedings of the $2^{nd}$ ICML Workshop on Foundation Models for Structured Data*, Seoul, South Korea. 2026. Copyright 2026 by the author(s).

sian, kNN); 4) quantifiable inspectability control through perplexity measurement and kernel scale tuning, enabling practitioners to navigate an accuracy-inspectability tradeoff. Figure 1 illustrates our approach.

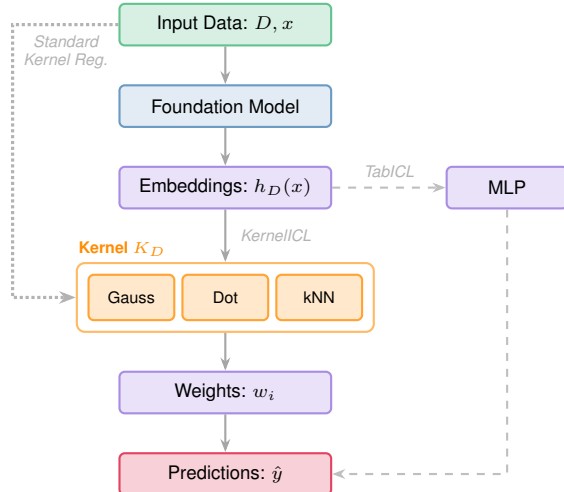

*Figure 1.* Three approaches to tabular prediction (see Section 2 for notation). **Standard Kernel Regression (dotted):** Kernel functions applied to inputs yield transparent predictions but lack learned representations. **TabICL or similar (dashed):** Foundation model learns powerful embeddings but uses an opaque MLP head. **KernelICL (ours, solid):** Combines both by fine-tuning the foundation model with explicit kernel form, producing transparent weighted averages with inspectable coefficients.

## 2. Methodology

Given a context dataset $D = \{(x_i, y_i)\}_{i=1}^n$, we aim to predict the label for a test sample $x$. Standard tabular foundation models pass both $D$ and $x$ through a transformer to obtain embeddings, which are fed into an opaque MLP head. KernelICL replaces this MLP with a classical Nadaraya-Watson estimator (Nadaraya, 1964; Watson, 1964):

$$\hat{y}(x) = \sum_{i=1}^n w_i y_i = \frac{\sum_{i=1}^n K_D(q_D(x), k_D(x_i)) y_i}{\sum_{j=1}^n K_D(q_D(x), k_D(x_j))}$$

where $q_D$ and $k_D$ embed the test and training samples into query and key spaces, and $K_D$ is a kernel measuring similarity.

To achieve prediction-layer inspectability, we constrain the architecture to transition from an opaque MLP to a transparent, distance-based prediction head. For an extended discussion on the structural constraints of these prediction mechanisms, please refer to Appendix B

### 2.1. KernelICL: Symmetric In-Context Kernel Regression

For a distance-based kernel (like Gaussian or kNN) to be geometrically meaningful, the query and key spaces must be

unified ($q_D = k_D$). However, standard in-context learning is inherently asymmetric: training samples act as context (keys/values), while test samples act as queries. KernelICL achieves symmetric embeddings by exploiting TabICL's (Qu et al., 2025) *staged* architecture.

Schematically, in-context embedding $E$ for a dataset with training samples $X_{\text{train}} = \{x_i\}_{i=1}^n$, training labels $y_{\text{train}} = \{y_i\}_{i=1}^n$ and test samples $X_{\text{test}} = \{x_j\}_{j=1}^m$ operates as:

$$E_{\text{train}}, E_{\text{test}} = \text{TF}((X_{\text{train}}, y_{\text{train}}), X_{\text{test}}) \qquad (1)$$

where TF is a transformer with attention masking that prevents training samples from attending to test samples. Since training and test samples have distinct roles (context vs query), their embeddings differ even for identical inputs. To obtain symmetric embeddings, one would need to pass all training samples as both context and queries, effectively doubling the computational cost.

Our approach leverages TabICL's (Qu et al., 2025) three-stage embedding architecture, which separates column-wise feature processing, row-wise sample interaction, and label-conditioned in-context learning:

$$E_{\text{train}}^c, E_{\text{test}}^c = \text{TF}_{\text{col}}(X_{\text{train}}, X_{\text{test}}) \qquad (2)$$
$$E_{\text{train}}^{cr}, E_{\text{test}}^{cr} = \text{TF}_{\text{row}}(E_{\text{train}}^c, E_{\text{test}}^c) \qquad (3)$$
$$E_{\text{train}}, E_{\text{test}} = \text{TF}_{\text{icl}}(E_{\text{train}}^{cr} + g(y_{\text{train}}), E_{\text{test}}^{cr}) \qquad (4)$$

where $g$ encodes training labels. Crucially, $\text{TF}_{\text{col}}$ processes each feature column via Set Transformers (Lee et al., 2019) with learnable inducing vectors that attend only to training samples, computing distributional statistics broadcast to all positions. This makes $\text{TF}_{\text{col}}$ apply identical operations to all samples (train and test) for a given training set. Similarly, $\text{TF}_{\text{row}}$ performs feature-wise self-attention within each row, again position-agnostic. Only $\text{TF}_{\text{icl}}$ introduces asymmetry via distinct attention masks for context versus query.

To achieve symmetric embeddings, we reprocess training samples as additional queries through $\text{TF}_{\text{icl}}$:

$$\_, E_{\text{train}} \parallel E_{\text{test}} = \text{TF}_{\text{icl}}(E_{\text{train}}^{cr} + g(y_{\text{train}}), E_{\text{train}}^{cr} \parallel E_{\text{test}}^{cr}) \quad (5)$$

where $\parallel$ denotes concatenation along the sample axis and the underscore indicates the discarded first output (context embeddings). Since $\text{TF}_{\text{col}}$ and $\text{TF}_{\text{row}}$ already apply identical operations, only $\text{TF}_{\text{icl}}$ requires reprocessing and incurs an overhead compared to typical asymmetric ICL embeddings.

While TabICL passes its 512-dimensional test embeddings $E_{\text{test}}$ to an MLP for predictions, we instead apply a learnable projection $W \in \mathbb{R}^{512 \times d_k}$ to define embedding functions:

$$k_D(x_i) = [W \cdot E_{\text{train}}]_i, \quad q_D(x_j) = [W \cdot E_{\text{test}}]_j \qquad (6)$$

for each training sample $x_i$ and test sample $x_j$. These embeddings are passed to a kernel function $K_\gamma$ (explored in

Section 2.2) to compute kernel weights. By construction, identical inputs produce identical projected embeddings regardless of whether they appeared in context or query positions, achieving $q_D = k_D = h_D$ as required by Equation (13) for geometric inspectability.

## 2.2. Kernel Function Exploration

The decomposition $\kappa_D(x, x_i) = K_D(h_D(x), h_D(x_i))$ from Section B separates learned geometry (embedding $h_D$) from similarity measurement (kernel $K_D$). Using simple, single-parameter kernels for transparent predictions concentrates representational complexity in the embedding function, while keeping the kernel operation transparent and inspectable. Among kernel families, distance-based kernels offer particularly intuitive notions of similarity: samples are near or far in the learned space. However, distance-based kernels require symmetric embeddings to have geometric meaning. KernelICL's symmetric mode (Section 2.1) enables systematic exploration of distance-based kernels in the ICL setting. We examine three specifications of the kernel function $K_\gamma(q, k)$ for $q \in \mathcal{Q}, k \in \mathcal{K}$ spanning the inspectability spectrum.

**Dot-Product Kernel.** Standard transformer attention (Vaswani et al., 2017) uses the exponential dot-product kernel:

$$K_\gamma^{\text{Dot}}(q, k) = \exp\left(\gamma\, q^T k\right) \qquad (7)$$

where $\gamma$ is a scale parameter controlling the sharpness of the distribution. The default scale is $\gamma = 1/\sqrt{d_k}$, a standard choice in attention mechanisms to stabilize the variance (Vaswani et al., 2017). The dot-product kernel works in both asymmetric and symmetric modes, though only symmetric embeddings enable geometric interpretation where weights reflect similarity in a shared space. SoftKNN-ICL (Koshil et al., 2025) used this kernel asymmetrically for ICL. The dot-product kernel measures similarity through alignment rather than distance.

**Gaussian Kernel.** Symmetric embeddings enable distance-based kernels for in-context learning. The Gaussian kernel

$$K_\gamma^{\text{Gaussian}}(q, k) = \exp(-\gamma\|q - k\|^2) \qquad (8)$$

controls locality via $\gamma$: large $\gamma$ concentrates weight on nearby samples, small $\gamma$ distributes weight uniformly. We use default $\gamma = 1/(2\sqrt{d_k})$; under unit-norm embeddings, this makes the Gaussian kernel equivalent to the dot-product kernel as $\|q - k\|^2 = 2 - 2q^T k$. Unlike the dot-product kernel, the Gaussian kernel is isotropic, depending only on Euclidean distance, offering intuitive spatial interpretation of similarity.

**kNN Kernel.** With symmetric embeddings providing a shared geometric space, kNN becomes applicable to in-

context learning. The kNN kernel is defined as:

$$[K_\gamma^{\text{kNN}}(q, \boldsymbol{k})]_i = \begin{cases} 1 & \text{if } \|q - k_i\| \leq \sigma_\gamma(q, \boldsymbol{k}) \\ 0 & \text{otherwise,} \end{cases} \qquad (9)$$

where $\boldsymbol{k} = (k_1, \ldots, k_n)$ denotes all training keys and $\sigma_\gamma(q, \boldsymbol{k})$ is the $\gamma$-th smallest distance among $\{\|q - k_i\|\}_{i=1}^n$. The scale $\gamma$ (or $k$ when there is no ambiguity with the key) represents the number of neighbors. Our kernel regression approach naturally extends to such vectorial kernels, though we use pairwise kernels $K_\gamma(q, k_i)$ in the exposition for simplicity. The kNN kernel's binary weights provide maximum inspectability: predictions use exactly $\gamma$ samples, though at the cost of differentiability due to the sorting operation.

A visual illustration of the KernelICL approach and a distance-based weight concentration is given in Appendix E on a synthetic 2-dimensional dataset.

## 3. Experiments

### 3.1. Training and Synthetic Validation

We fine-tune the TabICL embedding module and the projection matrix $W$ end-to-end with cross-entropy loss on the kernel predictions using 5,000 batches of synthetic data from TabICL's prior distribution (64 datasets per batch). For the kNN kernel, which is non-differentiable due to neighbor selection, we use embeddings trained with the Gaussian kernel as they share the same distance-based structure. Note that there are also differentiable relaxations of top-k ranking operations (Swezey et al., 2021), which would enable end-to-end kNN training. Training details in Appendix C.

### 3.2. Benchmark Evaluation

We evaluate KernelICL's accuracy and inspectability on the 55 binary classification datasets in the TALENT benchmark (Liu et al., 2024; Ye et al., 2024). The details on the experimental setup, including the full list of datasets and baselines, are provided in Appendix F.

| Method | Mean Rank | Mean Accuracy (%) |
|---|---|---|
| TabICL (ensemble) | 4.95 | 83.33 |
| TabICL (single) | 5.52 | 83.05 |
| KernelICL-Gaussian | 6.25 | 82.87 |
| TabICL-MLP | 6.39 | 82.91 |
| KernelICL-Dot | 6.49 | 82.88 |
| KernelICL-kNN | 6.75 | 82.79 |

*Table 2.* Subset of benchmark results for methods using the TabICL embedding architecture. KernelICL variants achieve similar accuracy to TabICL-MLP while providing inspectable weights.

Figure 2 presents a statistical comparison across 14 methods on 55 TALENT datasets. The critical difference diagram

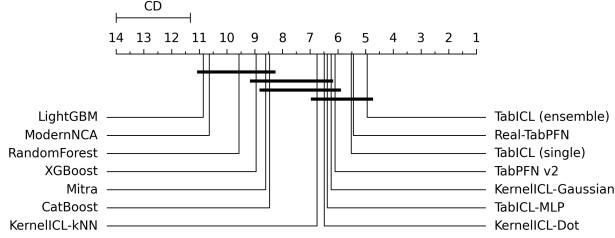

*Figure 2.* Critical difference diagram comparing 14 methods on 55 TALENT binary classification datasets. Methods connected by horizontal bars show no statistically significant difference in accuracy. KernelICL variants form a tight cluster with TabICL and TabPFN, demonstrating that explicit kernel constraints preserve competitive performance.

indicates no statistically significant performance difference between KernelICL variants and TabICL or TabPFN variants. While TabICL (ensemble) achieves superior mean rank (4.95), all KernelICL variants (ranks 6.25-6.75) fall within the critical difference threshold (2.68), showing that explicit kernel constraints preserve competitive performance. Table 6 (Appendix) provides complete rankings.

Table 2 examines methods sharing the TabICL embedding architecture. TabICL-MLP provides a controlled comparison: a similar architecture to TabICL (embedding + MLP) but fine-tuned with the KernelICL procedure. All three KernelICL variants match TabICL-MLP within 0.12 accuracy points, demonstrating that transparency through explicit kernel form comes at negligible accuracy cost.

### 3.3. Inspectability-Accuracy Trade-off

A transparent prediction is only practically inspectable if it relies on a human-manageable number of training samples. We quantify this using the perplexity of the weight vector: $\text{PPL}(w) = \exp(-\sum_{i=1}^{n} w_i \log w_i)$. Lower perplexity indicates sparser weights (e.g., $\text{PPL}(w) = 5$ effectively acts as a 5-nearest neighbors prediction). To compare across varying dataset sizes, we report the geometric mean of the *relative perplexity* ($\text{PPL}(w)/n$) across all test points.

While cross-validated scale selection (detailed in Appendix G) maximizes accuracy within the mechanistic framework, the resulting 11-29% relative perplexity may not provide adequate inspectability. Depending on the dataset size, 10% relative perplexity still corresponds to many samples being examined. KernelICL enables practitioners to adjust the kernel scale to achieve their desired sparsity level when inspecting predictions, accepting an accuracy cost for inspectability.

Figure 3 shows the resulting trade-off. For a hyperparameter grid (bandwidth $\gamma$ or neighborhood size $k$), we measure accuracy and relative perplexity on the test set. The x-axis

shows target relative perplexity levels; for each target, we select the hyperparameter with perplexity closest to (but not exceeding) the target, and report its accuracy averaged across the 55 datasets. At 100% perplexity, weights are uniform and methods converge to the baseline ($\sim 70\%$).

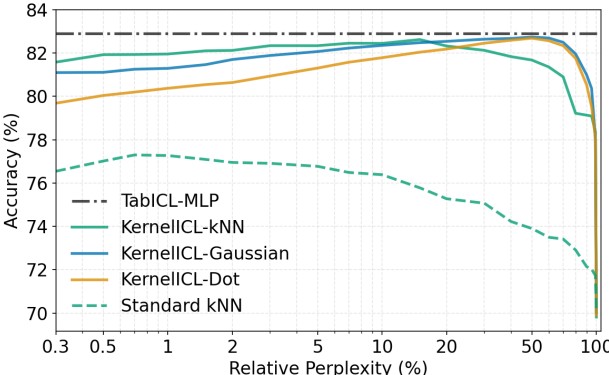

*Figure 3.* Accuracy-sparsity trade-off for KernelICL variants and standard kNN. TabICL-MLP shown for reference.

At comparable sparsity, KernelICL-kNN achieves $\sim 5$ percentage points higher accuracy than standard kNN, confirming that learned embeddings are essential. To examine whether this improvement reflects meaningful similarity, we analyze neighbor selection on the Pima Indians Diabetes dataset. Measuring neighborhood compactness per feature, KernelICL-kNN shows tight neighborhoods on glucose (+61% vs standard kNN) and BMI (+35%), the primary risk factors in medical literature (DeFronzo et al., 2015), while standard kNN treats all features roughly equally. This suggests learned embeddings concentrate similarity along clinically relevant dimensions. Full analysis in Appendix D.

Below 10% perplexity, KernelICL-kNN achieves highest accuracy among KernelICL variants. Comparing soft kernels, Gaussian and Dot-product achieve similar peak accuracies (close to TabICL-MLP at 20 to 60% perplexity), but at lower perplexities (below 20%), Gaussian consistently outperforms Dot-product, indicating distance-based similarity suits sparse regimes better than alignment-based measures.

## 4. Conclusion

We introduced KernelICL, a framework that replaces the prediction head of tabular foundation models with multiple and explicit kernel functions. Every prediction becomes a weighted average of training labels with inspectable coefficients. Perplexity metrics quantify inspectability, enabling practitioners to explicitly control the Inspectability-Accuracy trade-off via the data-dependent scale. On 55 TALENT datasets, KernelICL achieves 82.88% accuracy, within 0.2% of TabICL, while providing explicit sample weights.

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

## A. Related Work

**Tabular Foundation Models.** Foundation models for tabular data have evolved rapidly, achieving remarkable performance through in-context learning. TabPFN (Hollmann et al., 2022) pioneered this direction by training a transformer on millions of synthetic datasets, enabling predictions on real datasets with up to 1,000 samples in under a second without hyperparameter tuning. TabPFNv2 (Hollmann et al., 2025) extended this to 10,000 samples through alternating column-wise and row-wise attention. TabPFN-2.5 (Grinsztajn et al., 2025) has extended scalability to 50,000 samples, achieving parity with complex four-hour tuned ensembles.

While these models excel on small to medium datasets, TabICL (Qu et al., 2025) addresses scalability to large data through a two-stage architecture: distribution-aware column-wise embeddings followed by context-aware row-wise interaction produce fixed-dimensional representations, which are then processed by a transformer for efficient ICL. This design enables handling up to 500,000 samples while maintaining competitive performance.

**Interpretable Tabular Learning.** The interpretability literature typically distinguishes between inherently interpretable models and post-hoc explanations of black-box predictions (Rudin, 2019). Post-hoc methods like SHAP (Lundberg & Lee, 2017) and LIME (Ribeiro et al., 2016) compute feature attributions, and while recent work has developed kernel-smoothing perspectives for such measures (Miftachov et al., 2025), these approaches may still be unfaithful to actual model reasoning (Rudin, 2019; Lipton, 2018). For high-stakes tabular decisions, models which are inherently interpretable by design are thus often preferable.

Among other inherently interpretable approaches, generalized additive models (GAMs) decompose predictions as sums of univariate functions (Hastie & Tibshirani, 1986; Fan et al., 1998). GAMformer (Mueller et al., 2024) introduced the first foundation model approach to GAMs, using ICL to estimate shape functions in a single forward pass. A complementary approach provides example-based explanations through explicit dependence on training samples, where classical kNN (Cover & Hart, 1967) is the canonical example. This aligns with case-based reasoning common in domains like medicine and law, where practitioners validate decisions by comparing to past cases (Molnar, 2020).

ModernNCA (Ye et al., 2025) combines kNN with deep learning, learning embeddings via a Neighbourhood Components Analysis objective with stochastic neighborhood sampling. It achieved state-of-the-art results but requires per-dataset model training. TabR (Gorishniy et al., 2023) combines learned embeddings with kNN retrieval. SoftKNN-ICL (Koshil et al., 2025) operates in the ICL setting, using

attention to produce weighted averages of training labels, but employs only a dot-product attention-like kernel.

**Kernel View of In-Context Learning.** Recent theoretical work has established connections between attention mechanisms and kernel methods. Han et al. (2025) show that in-context learning predictions can be asymptotically approximated by kernel regression as the number of context examples increases, and empirically validate this by reconstructing large language model predictions from attention weights with high accuracy. Their framework explains several ICL phenomena, including why retrieving similar demonstrations improves performance. However, this kernel structure emerges implicitly rather than being enforced by design.

The connection between attention and kernel regression is well-established. Standard scaled dot-product attention (Vaswani et al., 2017) can be viewed as Nadaraya-Watson estimation (Nadaraya, 1964; Watson, 1964) with an exponential kernel: $K(q, k) = \exp(q^\top k / \sqrt{d_k})$, where $q$ is the query, $k$ is the key, and $d_k$ is the key dimension. This equivalence reveals that the exponential dot-product kernel underlying scaled dot-product attention is one choice among many possible kernels (Genton, 2001), each encoding different notions of similarity and locality.

## B. Structural Constraints of Prediction Layers

Sample-based interpretability allows predictions to be understood through individual training examples, supporting case-based reasoning where practitioners validate decisions by examining similar past cases. This contrasts with feature-based interpretability (e.g., GAMformer (Mueller et al., 2024)), which decomposes predictions into feature contributions. We characterize the transparency of the prediction mechanism through progressively constrained forms.

**Level 0: Opaque.** In the most general form, predictions are computed by an arbitrary function:

$$\hat{y}(x) = f_D(x) \tag{10}$$

While $f_D$ may achieve strong performance, it provides no inherent structure for mechanistic understanding. Interpretability, if desired, must rely on post-hoc explanation methods.

**Level 1: Mechanistic.** A first constraint, following Nadaraya (1964); Watson (1964), requires predictions to take the form of weighted averages:

$$\hat{y}(x) = \sum_{i=1}^{n} w_i y_i = \frac{\sum_{i=1}^{n} \kappa_D(x, x_i) y_i}{\sum_{j=1}^{n} \kappa_D(x, x_j)} \tag{11}$$

where $\kappa_D : \mathcal{X} \times \mathcal{X} \to \mathbb{R}^+$ is a weighting function[1] that

---

[1] $\kappa_D$ need not satisfy standard kernel assumptions from non-

can depend on the dataset $D$. The prediction is now an explicit linear combination of training labels with inspectable coefficients $w_i$.

To further characterize interpretability, we decompose the weighting function. Without loss of generality:

$$\kappa_D(x, x_i) = K_D(q_D(x), k_D(x_i)) \tag{12}$$

where $q_D : \mathcal{X} \to \mathcal{Q}$ embeds the test sample into a query space, and $k_D : \mathcal{X} \to \mathcal{K}$ embeds training samples into a key space (each key retrieves an associated value $y_i$). The kernel $K_D : \mathcal{Q} \times \mathcal{K} \to \mathbb{R}^+$ then computes a scalar similarity measure. This decomposition separates learned geometric transformations (the embeddings) from a scalar summary (the kernel). When the query and key spaces differ, geometric interpretation is precluded.

**Level 2: Geometric.** A further constraint requires shared embedding functions: query and key spaces become unified. We denote this shared function by $h_D$, where $q_D = k_D = h_D$, mapping $\mathcal{X} \to \mathcal{H}$:

$$\kappa_D(x, x_i) = K_D(h_D(x), h_D(x_i)) \tag{13}$$

Now both test and training samples inhabit a common embedding space $\mathcal{H}$, where kernel weights $w_i \propto K_D(h_D(x), h_D(x_i))$ have geometric meaning: they measure similarity between $x$ and $x_i$ in the learned space. The weight distribution becomes a geometric snapshot from the test point's perspective.

**Level 3: Distance-Based.** The most constrained form uses an isotropic kernel (Genton, 2001), depending solely on Euclidean distance in embedding space:

$$K_D(h_D(x), h_D(x_i)) = F_D(\|h_D(x) - h_D(x_i)\|) \tag{14}$$

for some monotone decreasing function $F_D : \mathbb{R}^+ \to \mathbb{R}^+$. Distance-based kernels provide a concrete spatial mental model for the abstract notion of similarity: samples are near or far, offering an intuitive handle for understanding the weight distribution.

### B.1. Dataset Dependence Categories

Having established interpretability levels, we now characterize how model components adapt to each dataset $D$, progressing from fixed methods to increasingly adaptive approaches.

**Level 0: Fixed.** The simplest case uses fixed embeddings and a fixed kernel:

$$\kappa_D(x, x_i) = K(q(x), k(x_i)) \tag{15}$$

---

parametric statistics (Härdle, 1990), as we make no asymptotic consistency claims.

where both $q$ and $k$ are fixed functions (e.g., identity or polynomial features) and the kernel $K$ has no parameters that depend on the data. This is the standard Nadaraya-Watson estimator.

**Level 1: Scale Adaptation.** A first generalization allows a *scale* (or *bandwidth*) parameter $\gamma_D$ to adapt to each dataset:

$$\kappa_D(x, x_i) = K_{\gamma_D}(q(x), k(x_i)) \tag{16}$$

The embeddings $q$ and $k$ remain fixed, but a scale parameter $\gamma_D$ is tuned per dataset (e.g., via cross-validation). Different datasets have different intrinsic densities and scales; adapting $\gamma_D$ accounts for this without changing the underlying representation.

**Level 2: Geometric Adaptation.** The next generalization uses embeddings that depend on the dataset:

$$\kappa_D(x, x_i) = K_{\gamma_D}(q_D(x), k_D(x_i)) \tag{17}$$

Embeddings may depend arbitrarily on $D$, whether learned via in-context learning (foundation models) or per-dataset training (e.g., ModernNCA).

**Level 3: Kernel Selection.** The most general case selects kernel structure from a rich function space:

$$\kappa_D(x, x_i) = K_D^*(q_D(x), k_D(x_i)) \tag{18}$$

where $K_D^*$ is chosen per dataset from a parameterized family. Examples include kernel mixtures of different types ($K_D = \sum_j \alpha_j(D) K_j$ where $K_j$ can be Gaussian, Epanechnikov, etc.) (Gönen & Alpaydın, 2011), or learning multiple kernel shape parameters beyond a single scale (e.g. Silverman (1986)). Selection could be performed via cross-validation. This flexibility comes at the cost of interpretability: the kernel function itself becomes complex rather than remaining a transparent similarity measure.

## C. Training Details

We sample 5000 batches from TabICL's synthetic data prior, where each batch contains 64 datasets generated from random MLP functions. The number of features ranges from 5 to 100, the sequence length contains up to 1024 observations in total, with the training data being between 60% and 80%. This requires approximately 40GB of GPU memory. For evaluation of the training process, we sample additional 32 batches from the same distribution and evaluate the loss function regularly on the validation data. Thus, we are using $32 \times 64 = 2048$ datasets for validation. We choose the parameters corresponding to the smallest validation loss as the final model parameters.

## D. Case Study Details: Pima Indians Diabetes

Section 3.3 summarizes findings on the Pima Indians Diabetes dataset (Smith et al., 1988). Here we provide method-ological details and complete results.

We compare standard kNN (Euclidean distance in input space) with KernelICL-kNN (distance in learned embedding space) using $k = 5$ neighbors on 154 test samples. KernelICL achieves 75.3% accuracy versus 68.8% for standard kNN.

For each test point, we compute the mean distance to its $k$ neighbors along each feature in standardized space. Table 3 reports values normalized by method mean, enabling comparison of relative feature emphasis. Medical literature establishes glucose and BMI as primary causal drivers of diabetes, with age as a secondary contributor (DeFronzo et al., 2015; Kahn et al., 2006).

*Table 3.* Neighborhood compactness on Pima Indians Diabetes (154 test samples, $k = 5$). Values normalized by method mean; positive relative difference indicates KernelICL has tighter neighborhoods.

| Feature | Standard | KernelICL | Rel. Diff. |
|---|---|---|---|
| Glucose | 1.21 | 0.59 | +61% |
| BMI | 1.17 | 0.81 | +35% |
| Age | 0.98 | 0.82 | +17% |
| BloodPressure | 1.07 | 1.02 | +6% |
| Pregnancies | 1.00 | 1.08 | −8% |
| DiabetesPedigree | 1.16 | 1.39 | −23% |
| Insulin | 0.71 | 1.11 | −41% |
| SkinThickness | 0.71 | 1.18 | −47% |

Standard kNN shows roughly isotropic behavior, while KernelICL concentrates on clinically established predictors. Whether this alignment reflects learned causal structure requires further clinical validation.

## E. Illustrative Example

Figure 4 illustrates the KernelICL approach using a distance-based kernel on a synthetic dataset. Kernel weights concentrate on samples nearby in the learned embedding space (middle panel), with concentration controlled by the scale parameter $\gamma$. This control matters because mechanistic interpretability alone does not ensure practical inspectability: with datasets containing thousands of samples, diffuse weight distributions become impractical for human examination. Practitioners can adjust $\gamma$ after training to achieve desired sparsity levels.

## F. Benchmark Details

**Experimental Setup.** We evaluate on 55 binary classification datasets from TALENT (Liu et al., 2024; Ye et al., 2024) using the standard 76%/24% train/test split. These datasets span various domains and range from 3 to 970 features and from 645 to 109,099 samples.

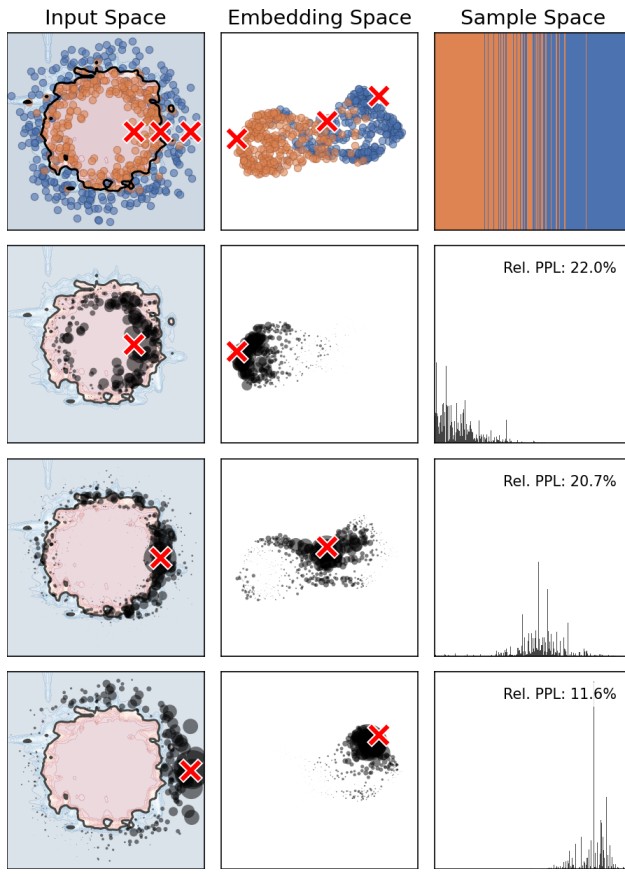

| Input Space | Embedding Space | Sample Space |

*Figure 4.* Illustration of the KernelICL approach with Gaussian kernel on a 2D synthetic dataset. **Left:** Input space $x$ showing concentric circles of different classes. **Middle:** 2D UMAP projection of 512D ICL embedding $h_D(x)$ showing class separation. **Right:** 1D "sample space", x-axis representing the training samples sorted by first UMAP dimension. **Top Row:** Training samples colored by class with decision boundary in input space. **Other Rows:** Weight $w_i$ (circle size for input and embedding space, height for sample space) of each training sample $i$ for 3 example test points (red crosses). Relative perplexity quantifies weight inspectability.

For KernelICL, we use symmetric mode with projection dimension $d_k = 512$ and perform 5-fold cross-validation on training data to select kernel scale $\gamma_D$ (calibration grids in Table 5) with $d_k = 512$ as embedding dimension.

**Metrics.** We report three metrics averaged over the 55 datasets: (1) mean accuracy (arithmetic mean), (2) mean rank (Wilcoxon signed-rank) by accuracy, and (3) mean time (average inference time per dataset in seconds). For KernelICL variants, we also report relative perplexity (arithmetic mean over the datasets of the geometric mean of $\mathrm{PPL}(w)/n$ across test samples within each dataset; defined in Section 3.2).

**Baseline Methods.** We compare against a number of baselines provided by the TALENT benchmark library[2]: founda-

---

[2]https://github.com/LAMDA-Tabular/TALENT

tion models (TabICL (Qu et al., 2025), Real-TabPFN (Garg et al., 2025), TabPFN v2 (Hollmann et al., 2025), Mitra (Zhang et al., 2025)), ModernNCA (Ye et al., 2025), and traditional machine learning methods (CatBoost, XG-Boost, RandomForest, LightGBM). All baselines use the defaults in the library. For ModernNCA, we use 20 epochs of training (no default). SoftKNN-ICL (Koshil et al., 2025) is not included as no public implementation is available; our KernelICL-Dot (non-symmetric, non-calibrated) configuration represents a similar approach with dot-product attention in asymmetric mode. For TabICL, the default is an ensemble method averaging predictions across 32 normalizations and feature/class shuffles. We add a non-ensemble (single) version for closer comparison with KernelICL which does not use ensembling for interpretability purposes. We also introduce a TabICL-MLP method using the same training procedure and architecture as KernelICL but replacing the projection $W$ and kernel with an MLP. TabICL-MLP is therefore very close to TabICL (single) in architecture but differs in its prior distribution due the effect of fine-tuning, thereby allowing to isolate the effects of the kernel regression head.

**Statistical Methodology.** We assess statistical significance using Friedman omnibus test followed by Nemenyi post-hoc test for pairwise comparisons (Demšar, 2006). Mean ranks are computed via Wilcoxon signed-rank test across the 55 datasets. The critical difference (CD) threshold determines when two methods are statistically indistinguishable; methods within CD are connected by horizontal bars in the CD diagrams.

## G. Ablation Studies

KernelICL's results in Section 3.2 use symmetric embeddings with $d_k = 512$ and calibrated kernel scales. We examine how those design choices impact accuracy, relative perplexity (inspectability), and runtime.

**Effect of Calibration.** Table 7 compares cross-validated scale calibration against default values. Calibration improves accuracy for all kernels while reducing perplexity for Gaussian and Dot-product variants. For those soft kernels, the accuracy gain is not statistically significant and does not justify the 21-24x runtime overhead of calibration for most practical purposes. For the kNN variant, the default $k = 5$ (following Scikit-learn) achieves high inspectability with 0.28% relative perplexity, while calibration significantly increases accuracy at the cost of an increase in perplexity to 11.89%, though sparser than soft kernels.

The 50x runtime cost highly depends on the hyperparameter grid and an alternate view on scale calibration is presented in Section 3.3.

**Effect of Symmetric Embeddings.** Section 2.1 introduced

*Table 4.* The 55 binary classification datasets from the TALENT benchmark (Ye et al., 2024) used for benchmarking. $N$ denotes the total sample size and $d$ denotes the number of features.

| Dataset | $N$ | $d$ | Train | Test | Dataset | $N$ | $d$ | Train | Test |
|---|---|---|---|---|---|---|---|---|---|
| Pima Indians Diabetes | 645 | 8 | 491 | 154 | Ada Agnostic | 3,832 | 48 | 2,919 | 913 |
| Sports Articles (Obj.) | 840 | 59 | 640 | 200 | Employee | 3,908 | 8 | 2,977 | 931 |
| Statlog | 840 | 20 | 640 | 200 | Wilt | 4,049 | 5 | 3,084 | 965 |
| QSAR Biodegradation | 885 | 41 | 674 | 211 | Company Bankruptcy | 5,728 | 95 | 4,364 | 1,364 |
| Golf Play (Extended) | 919 | 9 | 700 | 219 | Taiwanese Bankruptcy | 5,728 | 95 | 4,364 | 1,364 |
| PC1 | 931 | 21 | 709 | 222 | Water Quality | 6,716 | 20 | 5,116 | 1,600 |
| Diabetic Retinopathy | 967 | 19 | 736 | 231 | Bank Customer Churn | 8,400 | 10 | 6,400 | 2,000 |
| Basketball | 1,125 | 11 | 857 | 268 | JM1 | 9,143 | 21 | 6,966 | 2,177 |
| Banknote Auth. | 1,152 | 4 | 877 | 275 | E-Commerce Shipping | 9,239 | 10 | 7,039 | 2,200 |
| PC4 | 1,224 | 37 | 932 | 292 | Online Shoppers | 10,357 | 14 | 7,891 | 2,466 |
| IBM HR Analytics | 1,234 | 31 | 940 | 294 | Coupon Recommend. | 10,654 | 21 | 8,117 | 2,537 |
| Fitness Club | 1,260 | 6 | 960 | 300 | HTRU2 | 15,034 | 8 | 11,454 | 3,580 |
| PC3 | 1,313 | 37 | 1,000 | 313 | HR Analytics | 16,092 | 13 | 12,260 | 3,832 |
| Forex (AUD/JPY Day) | 1,539 | 10 | 1,172 | 367 | California Housing | 17,337 | 8 | 13,209 | 4,128 |
| Forex (AUD/CHF Day) | 1,539 | 10 | 1,172 | 367 | Android Permissions | 24,639 | 86 | 18,772 | 5,867 |
| Forex (AUD/CAD Day) | 1,540 | 10 | 1,173 | 367 | Default Credit Card | 25,200 | 23 | 19,200 | 6,000 |
| Forex (CAD/JPY Day) | 1,540 | 10 | 1,173 | 367 | INN Hotels Group | 30,471 | 17 | 23,216 | 7,255 |
| KC1 | 1,771 | 21 | 1,349 | 422 | Click Prediction (S) | 33,556 | 3 | 25,566 | 7,990 |
| Customer Personality | 1,881 | 24 | 1,433 | 448 | Forex (AUD/CAD Hour) | 36,813 | 10 | 28,048 | 8,765 |
| Marketing Campaign | 1,881 | 27 | 1,433 | 448 | Forex (AUD/JPY Hour) | 36,813 | 10 | 28,048 | 8,765 |
| NHANES | 1,913 | 7 | 1,457 | 456 | Forex (AUD/SGD Hour) | 36,813 | 10 | 28,048 | 8,765 |
| Pumpkin Seeds | 2,100 | 12 | 1,600 | 500 | Forex (AUD/USD Hour) | 36,813 | 10 | 28,048 | 8,765 |
| Wine | 2,145 | 4 | 1,634 | 511 | Forex (CAD/JPY Hour) | 36,813 | 10 | 28,048 | 8,765 |
| Seismic Bumps | 2,170 | 18 | 1,653 | 517 | Bank Marketing | 37,977 | 16 | 28,934 | 9,043 |
| Water Quality (Pot.) | 2,752 | 8 | 2,096 | 656 | Mobile C36 | 43,478 | 6 | 33,126 | 10,352 |
| Telecom Churn | 2,799 | 17 | 2,132 | 667 | Diabetes (130-US) | 85,483 | 20 | 65,129 | 20,354 |
| Gina Agnostic | 2,913 | 970 | 2,219 | 694 | Airline Satisfaction | 109,099 | 21 | 83,123 | 25,976 |
| Rice Cammeo | 3,200 | 7 | 2,438 | 762 | | | | | |

| Method | Hyperparameter Grid |
|---|---|
| KernelICL-Gaussian | $\gamma \in \{0.01, 0.05, 0.1, 0.3, 0.5, 0.8, 1, 3/2\}$ |
| KernelICL-Dot | $\gamma \in \{1/\sqrt{4}, 1/\sqrt{8}, 1/\sqrt{16}, 1/\sqrt{32},$ $1/\sqrt{64}, 1/\sqrt{128}, 1/\sqrt{256}, 1/\sqrt{512}\}$ |
| KernelICL-kNN | $k \in \{1, 4, 5, 16, 32, 64, 128, 256,$ $512, 1024, 2048, 4096, 8192\}$ |

*Table 5.* Scale calibration grids used for each KernelICL variant.

| Method | Rank | Accuracy (%) | Time (s) |
|---|---|---|---|
| TabICL (ensemble) | 4.95 | 83.33 | 2.9 |
| Real-TabPFN | 5.45 | 83.28 | 3.0 |
| TabICL (single) | 5.52 | 83.05 | 0.6 |
| TabPFN v2 | 6.10 | 83.16 | 2.9 |
| KernelICL-Gaussian | 6.25 | 82.87 | 42.9 |
| TabICL-MLP | 6.39 | 82.91 | 1.3 |
| KernelICL-Dot | 6.49 | 82.88 | 45.3 |
| KernelICL-kNN | 6.75 | 82.79 | 69.4 |
| CatBoost | 8.46 | 81.47 | 26.2 |
| Mitra | 8.61 | 82.31 | 5.3 |
| XGBoost | 8.95 | 80.99 | 0.2 |
| RandomForest | 9.57 | 80.70 | 32.6 |
| ModernNCA | 10.65 | 81.48 | 25.1 |
| LightGBM | 10.86 | 80.54 | 1.2 |

*Table 6.* Complete benchmark comparison on 55 TALENT datasets including foundation models, KernelICL variants, and traditional baselines. Metrics are arithmetic means over the datasets. Critical difference threshold on mean ranks: 2.68.

| Method | Acc. (%) | Perp. (%) | Time (s) |
|---|---|---|---|
| KernelICL-Gaussian | **82.87** | **28.63** | 42.9 |
| KernelICL-Gaussian (non-calibrated) | 82.81 | 37.35 | **2.0** |
| KernelICL-Dot | **82.88** | **28.81** | 45.3 |
| KernelICL-Dot (non-calibrated) | 82.79 | 38.64 | **1.9** |
| KernelICL-kNN | **82.79** | 11.89 | 69.4 |
| KernelICL-kNN (non-calibrated) | 81.28 | **0.28** | **1.4** |

*Table 7.* Effect of scale calibration on accuracy, relative perplexity, and runtime. Calibration uses 5-fold cross-validation, incurring computational overhead but improving both accuracy and perplexity. Best values per section and metric shown in bold.

symmetric embeddings where test and training samples are projected identically ($q_D = k_D = h_D$). We compare against a non-symmetric variant with separate projections for queries and keys ($q_D \neq k_D$) and without duplication of training samples, following SoftKNN-ICL (Koshil et al., 2025). Table 8 shows symmetric embeddings consistently achieve higher accuracy and lower perplexity across all kernels, with only 9% to 16% runtime overhead. The dual performance and inspectability gains justify the computational cost.

| Method | Acc. (%) | Perp. (%) | Time (s) |
|---|---|---|---|
| KernelICL-Gaussian | **82.87** | **28.63** | 42.9 |
| KernelICL-Gaussian (non-symmetric) | 82.44 | 40.59 | **39.3** |
| KernelICL-Dot | **82.88** | **28.81** | 45.3 |
| KernelICL-Dot (non-symmetric) | 82.79 | 47.35 | **39.2** |
| KernelICL-kNN | **82.79** | **11.89** | 69.4 |
| KernelICL-kNN (non-symmetric) | 82.52 | 16.57 | **60.4** |

*Table 8.* Effect of symmetric embeddings. Symmetric embeddings improve both accuracy and perplexity at minimal runtime overhead. Best values per section and metric shown in bold.

To better understand the overhead from symmetric embeddings, we measure embedding time on synthetic datasets with varying number of samples and features. Figure 5 shows the overhead reaches 100% in the large training set limit, with slower convergence at large number of features. The limit corresponds to $\text{TF}_{\text{icl}}$ dominating compared to the column-wise and row-wise embedding stages which do not need sample duplication. The methodology for this analysis is detailed in Appendix H.

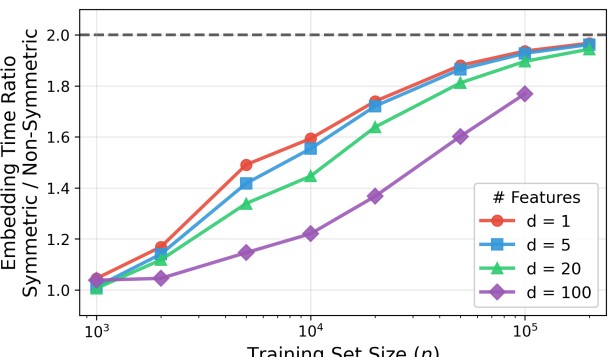

*Figure 5.* Overhead of symmetric embeddings on the embedding time measured on synthetic datasets, approaching a 2x factor in the large sample limit due to duplication of the training samples as both context and queries. Setups running out of memory are skipped.

**Effect of Projection Dimension.** Table 9 (Appendix) varies embedding dimension $d_k$ from 16 to 512. We use $d_k = 512$ for the main results as it provides the best combination of high accuracy and low perplexity.

Section G examined three design choices (scale calibration,

symmetric embeddings, projection dimension) measuring their impact on accuracy, perplexity, and runtime. Table 9 provides the projection dimension sweep, Table 10 extends analysis to all 23 ablation configurations including baselines, and Figure 6 shows statistical significance. Together, these results demonstrate that symmetric embeddings and calibrated scales consistently improve performance across all kernels.

| Method | Accuracy (%) | Rel. Perp. (%) |
|---|---|---|
| KernelICL-Gaussian ($d_k = 512$) | 82.87 | **28.63** |
| KernelICL-Gaussian ($d_k = 256$) | 82.88 | 32.00 |
| KernelICL-Gaussian ($d_k = 128$) | **82.91** | 35.20 |
| KernelICL-Gaussian ($d_k = 64$) | 82.89 | 40.83 |
| KernelICL-Gaussian ($d_k = 32$) | 82.68 | 37.18 |
| KernelICL-Gaussian ($d_k = 16$) | 82.87 | 37.50 |
| KernelICL-Dot ($d_k = 512$) | **82.88** | **28.81** |
| KernelICL-Dot ($d_k = 256$) | 82.78 | 35.39 |
| KernelICL-Dot ($d_k = 128$) | 82.70 | 38.86 |
| KernelICL-Dot ($d_k = 64$) | 82.79 | 40.63 |
| KernelICL-Dot ($d_k = 32$) | 82.63 | 57.61 |
| KernelICL-Dot ($d_k = 16$) | 82.79 | 60.12 |

*Table 9.* Effect of projection dimension $d_k$ on symmetric KernelICL with calibrated scale. Highest accuracy and lowest perplexity per section are shown in bold.

## H. Embedding Overhead

To isolate the computational overhead of symmetric embeddings from other inference components (I/O, preprocessing, cross-validation), we measure embedding time on synthetic datasets with controlled sizes. We generate binary classification datasets with $n \in \{10^3, 2 \times 10^3, 5 \times 10^3, 10^4, 2 \times 10^4, 5 \times 10^4, 10^5, 2 \times 10^5\}$ training samples, $m = 50$ (fixed) test samples and $d \in \{1, 5, 20, 100\}$ features, measuring the ratio of embedding times (symmetric / non-symmetric) on GPU (H100, 80GB memory). We only consider the TabICL embedding module, not including the projection to the $d_k$-dimension final embedding space since that projection has no overhead in symmetric mode.

Figure 5 shows the overhead ratio approaches 2× in the large sample limit, consistent with theory: symmetric mode processes training samples twice (once as context, once as query through $TF_{icl}$). Convergence is slower with more features because column-wise and row-wise stages ($TF_{col}$, $TF_{row}$) do not require sample duplication and represent larger fractions of total embedding time. The 2× overhead applies only to $TF_{icl}$, which becomes dominant for large $n$.

| Method | Mean Rank | Mean Accuracy (%) | Mean Time (s) | Rel. Perp. (%) |
|---|---|---|---|---|
| TabICL (ensemble) | 7.25 | 83.33 | 2.9 | - |
| TabICL (single) | 8.03 | 83.05 | 0.6 | - |
| Real-TabPFN | 8.56 | 83.28 | 3.0 | - |
| TabICL-MLP | 9.27 | 82.91 | 1.3 | - |
| KernelICL-Gaussian | 9.27 | 82.87 | 42.9 | 28.63 |
| TabPFN v2 | 9.44 | 83.16 | 2.9 | - |
| KernelICL-Dot | 9.75 | 82.88 | 45.3 | 28.81 |
| KernelICL-kNN | 10.11 | 82.79 | 69.4 | 11.89 |
| KernelICL-Dot (non-symmetric) | 10.29 | 82.79 | 39.2 | 47.35 |
| KernelICL-Dot (non-calibrated) | 10.41 | 82.79 | 1.9 | 38.64 |
| KernelICL-Gaussian (non-calibrated) | 10.65 | 82.81 | 2.0 | 37.35 |
| KernelICL-Dot (non-symmetric, non-calibrated) | 11.06 | 82.69 | 1.7 | 61.09 |
| KernelICL-kNN (non-symmetric) | 11.15 | 82.52 | 60.4 | 16.57 |
| KernelICL-Gaussian (non-symmetric) | 11.44 | 82.44 | 39.3 | 40.59 |
| KernelICL-Gaussian (non-symmetric, non-calibrated) | 12.05 | 82.50 | 1.8 | 51.43 |
| Mitra | 13.24 | 82.31 | 5.3 | - |
| CatBoost | 13.57 | 81.47 | 26.2 | - |
| XGBoost | 14.15 | 80.99 | 0.2 | - |
| RandomForest | 15.48 | 80.70 | 32.6 | - |
| KernelICL-kNN (non-calibrated) | 16.76 | 81.28 | 1.4 | 0.28 |
| ModernNCA | 16.92 | 81.48 | 25.1 | - |
| LightGBM | 17.61 | 80.54 | 1.2 | - |
| KernelICL-kNN (non-symmetric, non-calibrated) | 19.54 | 78.96 | 1.0 | 0.28 |

*Table 10.* Comprehensive ablation analysis on 55 TALENT datasets comparing 23 configurations: KernelICL variants (symmetric/non-symmetric embeddings, calibrated/non-calibrated scales), foundation models, and traditional baselines. Symmetric+calibrated KernelICL variants (ranks 9-10) cluster near top. Critical difference threshold on mean ranks: 4.68. Relative perplexity shown for KernelICL variants.

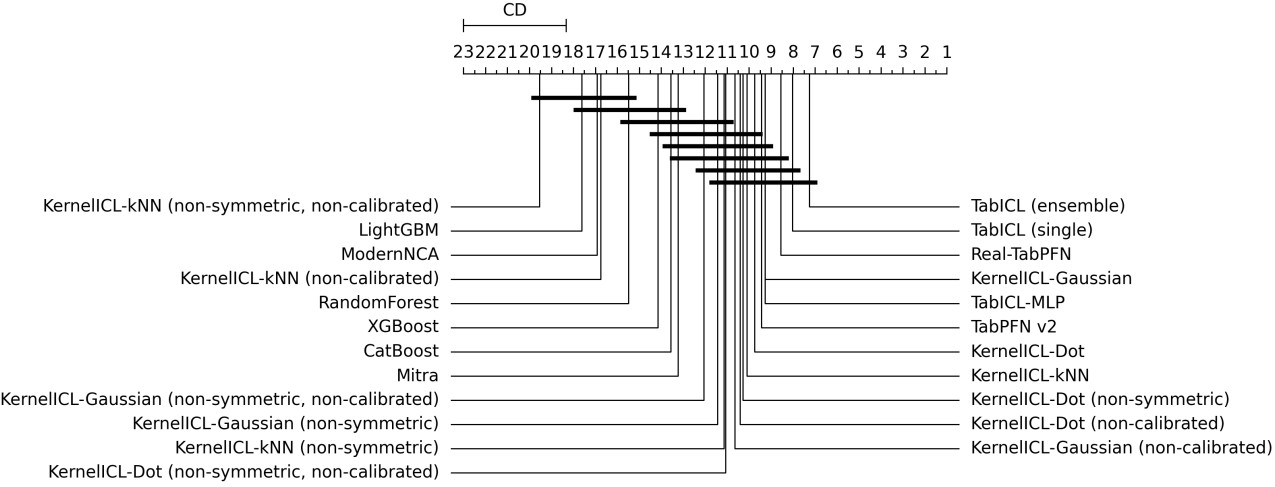

*Figure 6.* Critical difference diagram including KernelICL ablation configurations. Methods sharing horizontal bars show no statistically significant difference.

