# OpenReview forum: "Inspectable Tabular Foundation Models via In-Context Kernel Learning"
_ICML.cc/2026/Workshop/FMSD — FMSD @ ICML 2026 Poster_

### Official Review · Reviewer_Yexa · 2026-05-20

**Rating:** 7
**Confidence:** 4

**Review:**

## Summary

This paper proposes to modify Tabular Foundation Models to have a kernel prediction head at the top, so that test predictions are delivered as a function of an interpretable subset of instances from the training set. One key point of novelty is the generation of symmetric embeddings from the TabICL architecture (for the train and test sets), allowing the TabICL embedding pipeline to be repurposed for use in the resulting method, KernelICL. KernelICL is trained with the updated prediction head on TabICL priors, sacrificing only a small amount of performance.


## Strengths

- Good novelty on adding kernel layer for inspectability
- Intuitive architecture with good exposition
- Clever use of TabICL's embedding scheme
- Inspectability-Accuracy trade-off is interesting
- Resulting insights are interpretable

## Areas for Improvement

- Model is only inspectable on a sample level, which fundamentally limits what can be learned
- Querying $\text{TF}_\text{icl}$ again would be an expensive operation in the presence of larger data
- Relatedly, the method may really suffer in the regime where there are a very large number of test points, similar to other kernel-based approaches
  - Is it possible to consider inducing points?
- Need to incorporate more recent methods (e.g., TabICLv2 is > 3 months old, if pipeline is extensible it should be easy to make this modification)
- Clarity on training: from what I understand, the model is pre-trained with TabICL priors (appendix), but the main text mentions fine-tuning. Perhaps it is more like continued training of a TabICL model -- I think of fine-tuning as being done on a specific dataset (although maybe my interpretation is wrong). In any case, some clarity on training could be used in the main text.

## Detailed Comments

- What's the relationship with existing retrieval-based methods, such as LoCalPFN, TabDPT, and LimiX? How does KernelICL compare to those in terms of retrieved samples?
  - LoCalPFN is also worth noting as an early example of fine-tuning TFMs
- Can KernelICL easily be extended to regression considering TabICLv2 now supports it? Or is it less clear because TabICLv2 has a quantile head?
- What happens when you copy the training examples as test points? Does the kernel method only use the one relevant training point for each prediction?
- What happens if you cut off weights below a certain threshold and re-assign them as 0 weight? How much performance is sacrificed, and is it perhaps worth it for improved interpretability?


## Justification of Score

Solid workshop contribution, definitely thought-provoking and along a general line of work enabled by the fundamentally interesting properties of TFMs.

---

### Official Review · Reviewer_iCKF · 2026-05-22
**Interesting inspectable Tabular ICL framework, but interpretability claims need stronger validation**

**Rating:** 6
**Confidence:** 3

**Review:**

**Summary**

This paper proposes KernelICL, a framework for making tabular foundation models more inspectable by replacing the opaque MLP prediction head with explicit kernel-based prediction layers. The method represents each prediction as a weighted average of training labels, where the weights come from kernel functions such as Gaussian, dot-product, or kNN kernels. The authors introduce symmetric in-context embeddings so that distance-based kernels can be applied meaningfully in a shared embedding space. They evaluate KernelICL on 55 binary classification datasets from the TALENT benchmark and show that KernelICL variants achieve accuracy close to TabICL and TabPFN while providing sample-based weights for inspection.

**Strengths**

1. The paper addresses an important problem: improving inspectability of tabular foundation models. This is highly relevant for structured data applications in healthcare, finance, and other high-stakes settings.

2. The proposed framework is conceptually clear. Replacing the final prediction layer with kernel regression provides a transparent weighted-average prediction mechanism.

3. The use of symmetric embeddings is a useful technical idea. It enables distance-based kernels such as Gaussian and kNN to operate in a shared learned representation space.

4. The paper makes a useful distinction between accuracy and inspectability. The use of perplexity to quantify how many training samples effectively contribute to a prediction is a reasonable and interpretable metric.

**Areas for Improvement**

1. The inspectability claim needs stronger validation. Although KernelICL provides sample weights, it is not fully clear whether these weights produce explanations that are actually useful for practitioners.

2. The method improves prediction-layer transparency but does not make the learned embedding itself interpretable. Since the kernel operates in a learned 512-dimensional embedding space, the retrieved neighbors may still be difficult to interpret without additional explanation of the embedding geometry.

3. Runtime is a concern. KernelICL variants are much slower than TabICL single and TabICL ensemble in the reported benchmark. This weakens the practicality of the approach.

4. The accuracy differences are small, but KernelICL does not clearly improve predictive performance. The main benefit is inspectability, so the paper should more carefully justify that the added computational cost is worthwhile.

5. The benchmark focuses only on binary classification. It would be useful to evaluate whether the approach extends to multiclass classification and regression.

---

### Official Review · Reviewer_oWUZ · 2026-05-22

**Rating:** 7
**Confidence:** 3

**Review:**

The paper proposes KernelICL, which replaces the opaque MLP head in TabICL-style models with a kernel regression head. Predictions become a weighted average of training labels, making it possible to inspect which training examples influenced each output. The method studies dot-product, Gaussian, and kNN kernels, with symmetric embeddings so train and test samples are comparable in the same space.


Strengths : The motivation is clear and relevant: tabular foundation models are strong but hard to inspect. The method gives faithful prediction-layer explanations, since retrieved examples are actually used in the prediction rather than added post-hoc. The evaluation on 55 TALENT binary classification datasets shows KernelICL stays close to TabICL/TabICL-MLP in accuracy, with useful ablations on calibration, symmetry, and projection dimension.


Areas for Improvement : The method is only partially interpretable: the kernel head is transparent, but the learned embedding space remains opaque. Explanation usefulness is not strongly validated; Runtime is high for calibrated variants, and evaluation is mostly limited to binary classification, leaving regression, multiclass tasks, missing data, imbalance, and distribution shift open.

Comments : Use more precise wording such as sample-inspectable or prediction-layer transparent, rather than implying full interpretability. Add stronger explanation tests: neighbor stability, domain-expert evaluation, and comparisons with more methods like SHAP/LIME, IG, or LRP.  Discuss practical deployment issues more clearly, especially low-latency variants, privacy protections for exposed training rows, and whether the method is intended for real-time prediction or audit-time explanation.


The paper has a clear and interesting contribution: making the final prediction layer of tabular foundation models sample-inspectable.
The method is technically coherent: symmetric embeddings and explicit kernel heads all support the main idea. The main reasons for not giving a stronger score are limited explanation validation, high runtime, and the fact that the learned embedding space remains opaque.